



# Consumption of CH₃Cl, CH₃Br and CH₃I and emission of CHCl₃, CHBr₃ and CH₂Br₂ from a retreating Arctic glacier's forefield

Moya L. Macdonald[1], Jemma L. Wadham[1], Dickon Young[2], Chris R. Lunder[3], Ove Hermansen[3], Guillaume Lamarche-Gagnon[1], Simon O'Doherty[2]

[1] School of Geographical Sciences, University of Bristol, Bristol, BS8 1SS, UK
[2] School of Chemistry, University of Bristol, Bristol, BS8 1TS, UK
[3] Norwegian Institute for Air Research (NILU), Kjeller, NO-2027, Norway

*Correspondence to:* Moya L. Macdonald (m.macdonald@bristol.ac.uk)

**Abstract.**

The Arctic is one of the most rapidly warming regions of the Earth, with predicted temperature increases of 5 - 7 °C and the accompanying extensive retreat of Arctic glacial systems by 2100. This will reveal new proglacial land surfaces for microbial colonisation, ultimately succeeding to tundra over decades to centuries. An unexplored dimension to these changes is the impact upon the emission and consumption of halogenated organic compounds (halocarbons) from proglacial land surfaces. Halocarbons are involved in several important atmospheric processes, including ozone destruction, and despite considerable research, uncertainties remain in the natural cycles of some of these compounds. Using flux chambers, we measured halocarbon fluxes from proglacial land surfaces spanning recently-exposed sediments (<10 years), to approximately 1950 year old tundra in front of a High Arctic glacier. Proglacial land surfaces were found to consume methyl chloride ($CH_3Cl$) and methyl bromide ($CH_3Br$), with both consumption and emission of methyl iodide ($CH_3I$) observed. The largest consumption rates of these compounds occurred at the oldest, vegetated, tundra sites (-126 ±4, -1.8 ±0.04 and -0.13 ±0.03 nmol m⁻² d⁻¹, respectively for $CH_3Cl$, $CH_3Br$ and $CH_3I$). However, similar consumption rates were recorded at much younger sites with little soil development, but with the presence of extensive cyanobacterial mats (means of -106 ±7, -1.7 ±0.1, -0.01 ±0.03 nmol m⁻² d⁻¹ for $CH_3Cl$, $CH_3Br$ and $CH_3I$). Emission of chloroform ($CHCl_3$), bromoform ($CHBr_3$) and dibromomethane ($CH_2Br_2$) was detected across the forefield, with the highest emission of $CHCl_3$ from cyanobacterial mats (106 ±42 nmol m⁻² d⁻¹), $CHBr_3$ from bare sediment adjacent to the mats (0.7 ± 0.3 nmol $CHBr_3$ m⁻² d⁻¹) and $CH_2Br_2$ from the vegetated tundra (mean 0.8 ±0.3 nmol m⁻² d⁻¹). We have demonstrated that proglacial surfaces can consume and emit halocarbons despite their young age and low soil development. With future glacial retreat and the expansion of these surfaces, these fluxes may become more important in the future.



## 1 Introduction

Despite being present at only low concentrations in the atmosphere (part per trillion, ppt), halocarbons play an important role in the destruction of ozone by supplying halogens to the stratosphere and the troposphere (Butler, 2000; Mellouki et al., 1992; Montzka et al., 2011). Methyl chloride ($CH_3Cl$) and methyl bromide ($CH_3Br$) are the most important sources of chlorine (16%)

and bromine (50%) to the troposphere and are important contributors to stratospheric ozone loss (Carpenter et al., 2014). After $CH_3Cl$, chloroform ($CHCl_3$) is the next largest natural carrier of chlorine. Bromoform ($CHBr_3$) and dibromomethane ($CH_2Br_2$) are the most abundant short-lived brominated compounds and contribute ~ 4-35 % of bromine to the stratosphere (Montzka et al., 2011). Methyl iodide ($CH_3I$) is the most important very-short lived iodinated gas species in the atmosphere with a lifetime of ~ 7 days (Montzka et al., 2011). Some of the aforementioned gases have anthropogenic sources, many of which have reduced

in magnitude under the Montreal Protocol (Carpenter et al., 2014). This has increased the relative importance of the natural sources of these halocarbons. The contribution of halocarbons to atmospheric processes makes it important to fully constrain present day sources, and their likely change under future climate change scenarios.

Most natural sources of halocarbons involve biological processes driven by plants, algae and fungi, with methyl halides ($CH_3X$;

X = Cl, Br, I) generated as a by-product of methyltransferase activity and polyhalomethanes (e.g. $CHCl_3$, $CHBr_3$, $CH_2Br_2$) produced as a by-product of haloperoxidase activity (Manley, 2002). Marine biogenic sources are predominantly driven by macro- and micro-algae and are particularly important for $CHBr_3$ and $CH_2Br_2$ which are considered to be exclusively marine (Laturnus et al., 1998; Montzka et al., 2011; Sturges et al., 1993; Tokarczyk and Moore, 1994). The other halocarbons studied here ($CH_3X$, $CHCl_3$) also have a wide range of terrestrial biogenic sources, including tropical and temperate forests, temperate

peatlands and Arctic tundra (Farhan Ul Haque et al., 2017; Forczek et al., 2015; Rhew et al., 2008; Simmonds et al., 2010).

Although biological sources dominate, abiotic sources are also possible, including emissions from open oceans (Chuck et al., 2005; Stemmler et al., 2014), oxidation of soil organic matter and degradation of leaf litter and plants (Derendorp et al., 2012; Keppler et al., 2000; Wishkerman et al., 2008). Major, non-atmospheric, natural sinks of the halocarbons are the oceans

(primarily abiotic) and bacterial degradation in soils (Nadalig et al., 2014; Shorter et al., 1995; Ziska et al., 2013). The bacterial soil sink has been identified in wide ranging habitats from temperate forests to the tundra (e.g. Khan et al., 2012; Teh et al., 2009). Despite this considerable research, uncertainties remain around the magnitudes of natural sources and sinks of halocarbons due in part to large variation around mean fluxes caused by spatial and temporal variability (e.g. Dimmer et al., 2001; Leedham et al., 2013; Montzka and Reimann, 2011; Stemmler et al., 2014). Reduction of the uncertainties and increased

understanding of the processes influencing natural halocarbon fluxes are important for predicting future change.

A previously unstudied environment for halocarbon fluxes is the young soil found on the forefields of retreating glaciers. As the Arctic warms, increasing areas of land are being exposed by ongoing glacial retreat, a process that is forecast to continue





throughout the 21st century (ACIA, 2005; Graversen et al., 2008). The newly exposed sediment is colonised by microbes such as heterotrophic bacteria and fungi, $CO_2$- and nitrogen-fixing cyanobacteria and nitrogen-fixing diazotrophs who fix nutrients into the developing soil (Bradley et al., 2014; McCann et al., 2016). Soil stabilisation on newly exposed glacial forefields (i.e. prior to widespread plant colonisation) is primarily driven by cyanobacterial colonisation and the subsequent formation of soil

crusts (Hodkinson et al., 2003). Through nutrient-fixing and soil stabilisation processes, the microbial community enables the succession of higher plants, eventually leading to a tundra-type ecosystem for High Arctic locations (e.g. Hodkinson et al., 2003; Moreau et al., 2008).

Despite the forecasting of enhanced glacial retreat, trace gas emissions from the proglacial environment have not been well-

investigated with studies primarily focussing on $CO_2$ fluxes, particularly from higher plants on older surfaces, or $CH_4$ fluxes (Chiri et al., 2015; Muraoka et al., 2008). There have been no studies on halogenated trace gas fluxes from these environments and how they might be affected by the accelerated change occurring in the Arctic. With the expansion of proglacial soils through increasing glacial retreat in the coming decades, understanding the processes occurring in these soils is timely. To investigate the impact of soil development and the associated microbial to plant succession on halogenated trace gas fluxes,

we conducted in situ flux measurements of $CH_3Cl$, $CH_3Br$, $CH_3I$, $CHCl_3$, $CHBr_3$ and $CH_2Br_2$ at five sites spanning newly exposed soils (exposed <10 years ago) to established tundra (exposed approximately 1950 years ago) in front of a high Arctic glacier.

## 2 Study site

### 2.1 General description of the location

Midtre Lovenbreen is a small (5.4 km$^2$) valley glacier situated on the northern side of the Brøggerhalvøya Peninsula, in northwestern Svalbard (78º 53' N, 12º 04' E). The glacier has been in near-constant negative mass balance since measurements began in 1968, and probably since at least the 1930s (Kohler et al., 2007). Warming mean annual temperatures since the 1920s has resulted in approximately 1.1 km of glacial retreat from a prominent moraine to its current position 1.8 km from the fjord edge (Fig. 1). Between 1966 and 1990, this retreat resulted in the exposure of 2.3 km$^2$ of land, and is a process that continues

today (Moreau et al., 2008). The exposed area is characterised by the dominance of large rock fragments (> 5 cm ⌀) and is influenced by glacial runoff with intermittent and shifting meltwater channels. The progression of the community assemblages along the proglacial chronosequence has occurred at slower rates than are typical, with cyanobacterial crust and lichens still prevalent beyond 150 years of exposure (Hodkinson et al., 2003). Vascular plants and byrophytes are present sporadically, and increasingly, with exposure age. The area experiences a maritime polar climate. The mean air temperature at the weather

station in nearby Ny-Ålesund in July 2017, when this study was undertaken, was 6.1 ºC (Norway MET, 2017). Mean summer soil temperatures (~ 2mm below surface) on the proglacial foreland have been measured at 7-9 ºC (Hodkinson et al., 2003).



## 2.2 Specific descriptions of the sites

Five different land surface types were studied in four different locations along a transect between the glacial snout and the fjord (Fig. 1). The sites had different vegetation types and coverage (Fig. 2). The exposure ages of the sites (in years before 2017) were estimated from dates obtained by $^{14}$C dating and aerial photography in other studies (Hodkinson et al., 2003;

Moreau et al., 2008). The site nearest the glacier's snout (site *snout*) had an exposure age of approximately 5 years and was characterised by bare sediment, with little to no visible signs of life (Fig. 2a). Approximately 100 metres from the glacier's snout, the second site (site *pond mat*) was located on the margins of a dried-up (by July) snow-melt pond in a small depression between the moraines. Around the margins of the pond, cyanobacterial mats had begun to form (Fig. 2b). The surrounding moraines were still largely barren. The *pond mat* site is estimated to have been exposed for around 20 years. The third and

fourth sites were located near the middle of the transect on an expanse of relatively flat land behind (~south) the prominent Little Ice Age moraine (Fig. 1). Site *established mat* was located on the extensive cyanobacterial mats which cover large expanses of the flatter land (Fig. 2d). A site immediately adjacent to the mats where the mats had been disturbed by snow melt flowing from ponds (site *disturbed mat*) was also studied as a direct comparison (Fig. 2c). The exposure age of site *established mat* and *disturbed mat* was estimated at 100 years. The final site (site *tundra*) was located about 200 m from the coast (Fig.

2e). At this site, small bluffs of limestone and siltstone provided some shelter from the shifting nature of the glacial runoff rivers which otherwise hamper colonisation of much of the flood plain between the moraines and the fjord. Site *tundra* had a soil depth of about 15 cm and 100 % vegetation coverage. Dominant species included *Bryophyta* spp. and *Carex rupestris*, *Salix polaris* and *Racomitrium lanuginosum*. Radiocarbon dating near site *tundra* (~70 m west) has provided a date of exposure of 1850-1926 BP (Hodkinson et al., 2003).

## 3 Methods

### 3.1 Flux experiments

Four custom-made, cylindrical, Perspex flux chambers (0.029 m$^3$) composed of a collar (0.07 m height) and top (0.22 m height, Fig. 2f) were deployed for gas analysis between the 25$^{th}$ and 31$^{st}$ July 2017. Preliminary experiments were conducted near site *established mat* in 2016 to determine the impact on gas fluxes of covering the chambers with a reflective material so that the

experiments were conducted in the dark. The tests showed no statistical difference (2-sample t-test; Sect. 3.5) between covered and uncovered chambers (conducted in duplicates) for mean fluxes of CH$_3$Cl, CH$_3$Br and CH$_3$I (Fig. 3; other halocarbons not analysed, experiment conducted over 5 hours). Despite there being no statistical difference in gas concentration change, the covered chambers were used for the main experiments in 2017 to prevent over-heating when in direct sunlight, therefore minimising the influence of heat on the soil processes involved in the fluxes. The collar was embedded in the sediment surface

prior to sampling (at least 18 hrs) to allow gases released/ absorbed from breaking the surface to equilibrate with the background air concentrations. At site *tundra*, where plant roots were abundant, a small knife was used to cut through the roots





as the collar alone could not break through the surface. An integrated "trough" on the collar was filled with deionised water (14-18 MΩ-cm) to provide a leak-tight seal with the upper section of the chamber (Fig. 2f). A fan (24 $m^3$ $h^{-1}$; San Ace 60) was operated continuously during incubation to ensure the chamber air remained mixed. Tinytag temperature loggers (Gemini Data Loggers) were fixed to the underside of the chamber lid.

Two sampling ports, constructed from polypropylene BSP fittings, Luer-lock stop-cocks, and 20 cm polypropylene tubing (port A only, Fig. 2f), enabled gas sampling to be conducted 1 and 2 hours after sealing the chamber. Two types of gas sampling were conducted; first, 3.7 mL samples were taken for $CO_2$ and $CH_4$ analysis in the laboratory in Bristol, UK; second, 2.5 L samples were taken for halocarbon analysis by GCMS at the UK station in Svalbard. Sampling was conducted with four

replicates (four chambers). Each site was analysed on a different day, with sites *snout* and *pond-mat* analysed once (4 replicates), and sites *mat*, *disturbed mat*, and *tundra* analysed twice (two separate days of four replicates each, total of eight replicates). Chambers and collars were washed with deionised water and dried with paper towels between sites to minimise contamination.

Both a laboratory and a field blank test of the flux chamber equipment was conducted by placing the chambers onto aluminium-foil trays and filling the inside of the chamber collar with a 1 cm deep layer of de-ionised water to create a seal. For the field blank tests, the aluminium-foil trays were placed on wooden boards (to provide a flat surface) on the ground near to site *tundra*. The blank tests were conducted with four replicates and gases were measured as in Sect. 3.2 and 3.3.

**3.2 $CO_2$ and $CH_4$ sampling and analysis**

$CO_2$ and $CH_4$ were sampled in duplicate at each time point using a glass gas-tight syringe (Hamilton). Samples were taken from the ambient air (time 0) and from the chamber headspace via port B (Fig. 2f, time 1 and 2). 5.5 mL of air was drawn through the tap using the syringe and flushed to ambient prior to withdrawing a further 5.5 mL of sample into the syringe. 1.5 mL of the sample was used to flush a syringe filter (0.2 μm) and needle. The remaining 4 mL of sample was aseptically injected

into a 3.7 mL evacuated vial (Exetainer®; Labco) via the flushed 0.2 μm syringe-filter. Exetainers were stored (within 4 hours of sampling) and transported at +4 °C until analysis in the UK.

Exetainer samples were injected into an Agilent 7890A gas chromatograph (GC) fitted with a methaniser (at 395 °C) and an FID (flame ionising detector, at 300 °C). Separation of methane ($CH_4$) and carbon dioxide ($CO_2$) was achieved using a

molecular sieve 5A, 60-80 mesh, 8 ft x 1/8-inch column, held at 30 °C for 4 minutes, before being ramped at 50 °C per minute to 180 °C. Calibration standards (mixed air; BOC) were run twice daily. The percentage variance, limit of quantification and limit of detection for each gas is displayed in Table 1. Concentrations of the samples were calculated from a linear regression



line ($r$ >0.99, n=5) of manual dilutions of certified (± 5 %) standards with 5.0 grade Argon (BOC) fitted with an in-line gas desiccator. The Ideal Gas Law was used to convert gas concentrations to molar amounts which were then corrected for dilution.

### 3.3 Halocarbon sampling and analysis

2.5 L air samples for the analysis of halocarbons were taken using a small pump (SKC, Twin Port Pocket Pump) at 250 mL min$^{-1}$ into 3 L Tedlar gas tight bags (polypropylene fittings, SKC). All sample bags were flushed three times with synthetic zero air prior to use, with laboratory testing indicating this removed any background contamination. The length of sampling time (ten minutes) required the chambers to be sealed approximately twelve minutes apart to allow time for sampling. A sample of ambient air was taken between the sealing of the first and second chambers and again between the sealing of the third and

fourth chambers. An average of the mixing ratios of two ambient measurements was used as time 0 for the four chambers. Headspace analysis of each chamber was taken after 1 and 2 hours through the extended tubing of port A to further ensure mixing of the chamber air (Fig. 2f). A 3 L sample bag flushed and filled with synthetic air was connected to port B during sampling to maintain ambient pressure within the chamber and prevent air being drawn through the soil. 50 mL of chamber air was flushed through the port A tubing and the pump prior to taking the 2.5 L sample. Sample bags were kept in the dark

until analysis (within <20 hours) at the UK station in Ny-Ålesund.

Analysis of halocarbons with part per trillion (ppt) atmospheric concentration was conducted with a custom-built adsorption-desorption system (ADS; developed by the University of Bristol; Simmonds et al., 1995) connected to an automated gas chromatograph mass spectrometer (GCMS). 1.5 L of whole-air sample was drawn through a Nafion permeation drier

(continuous counter-purge of dry 5.0 ultra grade synthetic air at 170 mL min$^{-1}$) before being condensed onto an absorbent filled microtrap held at -50 ºC using electrical resistance (Peltier device). The concentrated sample was desorbed by raising the microtrap to 240 ºC using direct ohmic heating. The sample was carried through a fused silica transfer line (100 ºC) by 5.0 grade Helium, purified by a Universal Trap, into a Hewlett Packard 6890A Gas Chromatograph. Separation of methyl-chloride ($CH_3Cl$), methyl-bromide ($CH_3Br$), methyl-iodide ($CH_3I$), dibromomethane ($CH_2Br_2$), chloroform ($CHCl_3$) and bromoform

($CHBr_3$) was achieved using a 25 m capillary GC column (Varian, PoraBOND Q, 320 µm i.d., 5 µm film thickness) which was held at 40 ºC for 3 minutes, ramped at 22 ºC min$^{-1}$ to 84 ºC and held for 1 minute, then ramped at 22 ºC min$^{-1}$ to 250 ºC where it is held for 37.73 minutes (total time: 49 minutes). Samples were identified from their fragmentation spectra using a Hewlett Packard 5973 Mass Spectrometer Detector (quadrupole at 150 ºC, source at 230 ºC) scanning for selected ion masses (Table 1). Bromochloromethane ($CH_2BrCl$) and diiodomethane ($CH_2I_2$) were also scanned for (target ions of 128 and 268,

respectively; qualifier ions of 130 and 141, respectively). $CH_2BrCl$ was present in only trace amounts in the standard (below the limit of detection) and was thus not quantifiable. $CH_2BrCl$ is discussed in this manuscript based on relative changes to the peak area. $CH_2I_2$ was not present in the standard. This is likely due to its exceptionally short atmospheric lifetime (0.003 days; Law et al., 2006) meaning its highly unlikely to persist in the ambient atmosphere, from which the standard was made. $CH_2I_2$



was not detected during the experiments either, which follows with previous research that has only identified its production in marine environments, particularly by macroalgae and sea-ice microalgae (Carpenter et al., 2000, 2007).

Quantification of compounds was determined using GCWerks software (gcwerks.com) from the average peak area of the two

closest standard analyses, which were run every second sample. The standard was cryo-filled from the ambient air on 11[th] January 2017 at the Norwegian Zeppelin Observatory (operated by the Norwegian Institute for Air Research, NILU), 2 km south of Ny-Ålesund at 475 m a.s.l. on Zeppelin Mountain. The standard was calibrated on the Zeppelin Medusa (part of the Advanced Global Atmospheric Gases Experiment (AGAGE; Prinn et al., 2018)) using tertiary standards linked to the primary standards prepared at Scripps Institution of Oceanography (SIO) for $CH_3Cl$ and $CH_3Br$ (SIO-05 calibration scale), and for

$CHCl_3$ (SIO-98 calibration scale). $CH_3I$, $CHBr_3$ and $CH_2Br_2$ are calibrated via AGAGE tank comparisons carried out in Boulder, Colorado against National Oceanic and Atmospheric Administration (NOAA) calibration scales ($CH_3I$, NOAA-2004; $CHBr_3$, NOAA-2003; $CH_2Br_2$, NOAA-2003) using SIO tanks T-005B, T-009B and T-102B. Due to the increased number of steps to transfer these calibration scales, flux calculations for these species have an additional error associated with them. The detection limit (three times the baseline noise), limit of quantification (variance) and standard concentration for each

halocarbon is displayed in Table 1. The Ideal Gas Law was used to convert gas concentrations to molar amounts. The dilution from the synthetic air bag used to maintain ambient pressure during sampling was corrected for by accounting for the moles of gas removed during sampling at each time point. The results are presented as daily fluxes in nanomoles per metre squared of land surface per day (nmol m$^{-2}$ d$^{-1}$). Daily fluxes were calculated from the change in the number of moles of gas present in the headspace over the first hour of the experiment, corrected for the mean change in moles during the first hour of the field

blank tests. These mean blank changes were: 0.2 nmol $CH_3Cl$ m$^{-2}$, 0.01 nmol $CH_3Br$ m$^{-2}$, 0.003 nmol $CH_3I$ m$^{-2}$, -0.03 nmol $CHCl_3$ m$^{-2}$, -0.01 nmol $CHBr_3$ m$^{-2}$, -0.002 nmol $CH_2Br_2$ m$^{-2}$. Mean daily fluxes are presented ± 1 standard deviation. The daily fluxes were calculated from the change in moles in 1 hour because the majority of the 2 hour total change occurred within the first hour. For example, 78 to 90 % of the initial moles of $CH_3Cl$ and $CH_3Br$ present in the chamber were consumed within the first hour at sites *established mat* and *tundra*, with only 0.01 to 4 % of additional consumption in the second hour. For the

gases that were emitted, a similar pattern emerged where the proportion of gas emitted in the first hour of the total amount of gas emitted over the 2 hour experiment was an average of 59 % of $CHCl_3$, 61 % of $CHBr_3$ and 60 % of $CH_2Br_2$ at sites *established mat* and *tundra*. Presumably the slowdown in the rate of change after 1 hour was due to reactants being consumed from the air trapped inside the chamber. Because of this, we advocate that our daily flux rates (nmol m$^{-2}$ d$^{-1}$) are a minimum estimate.



### 3.4 Physical, chemical and biological sampling and analysis

### 3.4.1 In-field measurements and sampling

The internal chamber temperature was recorded at 5 minute intervals (Tinytag loggers; Gemini) and an average was calculated for the 2 hour duration of each experiment. At the end of the incubation, the chamber tops were carefully removed without

disturbing the sediment surface. Aliquots of sediment (~1 g) from the centre of each collar were taken aseptically using 15 mL sterile falcon tubes. These samples were frozen at -20 ºC within 4.5 hrs of sampling and were transported and stored at this temperature until analysis of cell numbers in Bristol within 55 days or less.

After the sterile samples were conducted, a soil moisture sensor (ML3 ThetaProbe, accuracy of ±1 %) was used to measure

the volumetric water content of the sediment in each quarter (0.03 m$^2$) of the chamber. Small cores (~ 4 cm deep) of the sediment were taken from the centre of two opposite quarters of the chambers' footprint. The cored samples were broken up and dried for 20 hours at 60 ºC prior to transport to the UK for soil texture, total carbon (TC) content, total nitrogen (TN) content, and organic matter (OM) content analyses

In the centre of each chamber, a corer was used to determine the depth of the water table. In some cases the water table could not be reached due to the presence of high numbers of large (> 5 cm diameter) rocks in the near sub-surface which were not practical to dig through.

### 3.4.2 Organic matter, total nitrogen, total carbon and soil texture

Prior to OM, TC and TN content and soil texture analyses, plant roots (present at site *tundra*) and pieces of cyanobacterial mat (present at site *established mat*) were removed with tweezers from the dried samples. Additionally, a sieve was used to remove small roots (> 2 mm) from the site *tundra* samples but it was not possible to remove roots smaller than this.

Samples for OM, TC and TN content analyses were re-dried at 105 ºC for 19 hours to ensure removal of water. Approximately

4 g of a known weight of the dried-sample from each quarter-chamber core was then furnaced at 450 ºC for 5 hours to determine the OM content (weight %) by mass loss on ignition. The larger weight of sample used here meant that some very small roots were likely present in these samples and may inflate the values. In comparison, TC and TN content was analysed on less than 20 mg of sample meaning no root matter was likely to be present.

An Elemental Analyser 1110 fitted with a TCD (temperature controlled detector) was used to measure percentage weight of TC and TN in an 8 to 19 mg, < 250 μm, well-mixed aliquot of the re-dried core sample by flash heating to 1000 ºC. TC and TN contents were quantified using a certified Aspartic acid standard containing 36.14 % C and 10.49 % N. This method has



an LOD of 0.01 % for both TC and TN and a precision of 0.06 % for TC and 0.01 % for TN (n=6) as determined from a soil standard containing 2.29 % TC and 0.21 % TN.

To determine the heterogeneity and average size of grains at each site, the remaining approximately 10 g of re-dried core
sample was sieved to determine the percentage weight of the sample with grain sizes greater and smaller than 2 mm.

### 3.4.3 Bacterial abundance

Counts of bacteria were conducted after methodology detailed by Bradley et al., (2016). Briefly, upon analysis the samples were defrosted and 100 mg sub-sampled into sterile microcentrifuge tubes (1.5 mL, Eppendorf), upon analysis. The sample
was diluted with 932 µL of Milli-Q (MQ) water (0.2 µm filtered) and fixed in 68 µL of 0.2 µm filtered 37 % formaldehyde (final concentration of 2.5 %). Samples were vortexed for 10 seconds (s) and sonicated for 1 minute at 30 ºC to disaggregate soil particles and separate the cells from them. The sample was then vortexed for 3 s with 10 µL of fluorochrome DAPI (4',6-diamidino-2 phenylindole) prior to being incubated for 10 minutes in the dark. The stained sample was vortexed for 10 s and 100 µL of this was filtered through a black Polycarbonate filter paper (0.2 µm pore size, 25 mm diameter) and then rinsed with
250 µL of MQ water (0.2 µm filtered). Bacterial cells were counted under UV light at 1000 X magnification using an Olympus BX41 microscope. MQ water (0.2 µm filtered) was used to wash the filtering apparatus between each sample. Blank controls, to which no soil or sediment was added, were dispersed throughout the samples. Ten random grids (each $10^3$ $\mu m^2$) were counted per sample. The number of cells per gram of wet weight sample was calculated. Cell numbers for the blank controls were below 50 cells $mL^{-1}$.

### 3.5 Statistical analysis

Differences between mean halocarbon fluxes from different sites were determined at the 95% confidence level (p-values < 0.05) using pair-wise Welch two sample t-tests conducted in R (version 3.02.1, 2015). Correlations between halocarbon fluxes and the physical, chemical and biological variables are estimated and presented using the "corrplot" package in R (Wei and
Simko, 2017). An average value per chamber was calculated for the physical and chemical variables where multiple analyses were conducted at each chamber (OM, TC, TN and texture; n=2). Matrices were produced from the data for all sites combined and from the data for three individual sites: *disturbed mat*, *established mat* and *tundra*. The individual site matrices were generated because of the disparity in land surface "type" between sites which results in large variation in physical, chemical and biological variables. Bacterial cell numbers were excluded as a variable for the "within site" correlation matrices because
the four measurements conducted per site were deemed too few to be included in the analysis. Similarly, matrices were not produced for sites *snout* and *pond-mat* which only had four halocarbon flux data points each.





### 3.6 Calculation of regional fluxes

### 3.6.1 Calculation of total proglacial fluxes in the Arctic

To determine if halocarbon fluxes from glacial forefields were important regionally, we calculated an Arctic proglacial total

flux. First, we assumed an averaged flux for each halocarbon across the Midtre Lovenbreen forefield by subdividing the land surface into thirds. The first third is represented by fluxes from sites *snout* and *pond-mat*, the middle by fluxes from sites *disturbed* and *established mat*, and the final third by fluxes from site *tundra*. This gave an average forefield flux of -62 nmol $CH_3Cl$ m$^{-2}$ d$^{-1}$, -1.0 nmol $CH_3Br$ m$^{-2}$ d$^{-1}$, -0.04 nmol $CH_3I$ m$^{-2}$ d$^{-1}$, 56 nmol $CHCl_3$ m$^{-2}$ d$^{-1}$, 0.5 nmol $CHBr_3$ m$^{-2}$ d$^{-1}$ and 0.4 nmol $CH_2Br_2$ m$^{-2}$ d$^{-1}$. The total area of proglacial land surface across the region has not been measured. Therefore, we assume that

the size of Midtre Lovenbreen's forefield (2.7 x 10$^6$ m$^2$) is representative and combine this area with an estimated 9996 land terminating glaciers (minimum elevation > 50 m above sea level) located above 60ºN (WGMS, 2012), to calculate a total Arctic proglacial land surface area of 2.7 x 10$^{10}$ m$^2$. The estimated Arctic proglacial land surface area was combined with the average proglacial halocarbon fluxes and an assumed growing season of 100 days (with negligible fluxes out with this time) to calculate the regional source and sink of each halocarbon in moles and tonnes per year. The growing season length of 100

days was determined as the approximate average number of days with no ground snow-cover (as determined by others e.g. Bekku et al. (2003)) measured at Ny-Ålesund weather station from 2009-2017 (102 ± 26 days; Gjelten, 2018). We assume that the net flux of all gases is zero when outside of the growing season due to snow-cover, low light (including no light during polar night) and low temperatures which would inhibit or reduce the rate of consumption or production processes in the soils to negligible or near-negligible rates. This would follow results from studies on other gas fluxes from soils during winter, e.g.

$CO_2$ consumption was determined to be 1-2 orders of magnitude lower in winter than in summer in Alaskan tundra (Welker et al., 2000). However, the confirmation of halocarbon fluxes outside of the growing season cannot be definitively determined without further field studies.

### 3.6.2 Calculation of Arctic tundra fluxes

For the halocarbons ($CHBr_3$ and $CH_2Br_2$) that have not been measured on tundra before, we calculate an Arctic tundra flux based on calculations by Rhew et al. (2007) as follows. We assume that the growing season lasts 100 days (with negligible fluxes out with this time, see Sect 3.6.1) and that the area of the Arctic tundra is 7.3 x 10$^{12}$ m$^2$ (Matthews, 1983). By assuming our site *tundra* fluxes are broadly representative of tundra as a whole, the average fluxes of $CH_2Br_2$ and $CHBr_3$ measured at site *tundra* in nmol m$^{-2}$ d$^{-1}$ are combined with the Arctic tundra area and growing season length to calculate an annual Arctic

flux in moles of gas per year, which was converted to Gigagrams of gas per year.



## 4 Results

### 4.1 Physical, chemical and biological differences between sites

The environmental context for the halocarbon fluxes measured here was provided by the inter- and intra-site variation of the following physical, chemical and biological parameters (Fig. 4). Volumetric water content and water table depth both varied

between and within sites with highest water content at site *tundra* (50 % v/v) but shallowest water tables at site *disturbed mat* (Fig. 4 a-b). The texture of the sediment in the top 5 cm at the sites illustrated the heterogeneity of the moraine and fluvial outwash landscape, with near 100 % grains < 2 mm ∅ representing low-energy and sheltered environments at sites *tundra* and *pond-mat* compared to more variation at the other three sites (Fig. 4c).

The chemical and biological parameters describe the increasing soil development with distance from the glacier's snout, and therefore with exposure age. For example, bacterial cell abundances increased with distance from the glacial snout, with highest mean abundances at sites *established mat* and *tundra* of $3.2 \times 10^8$ cells [g sediment]$^{-1}$, compared with $0.6 \times 10^8$ cells [g sediment]$^{-1}$ at site *snout* (Fig. 4g). The highest soil contents of OM, TC and TN were all measured at site *tundra* (Fig. 4d-f). Net emission of $CO_2$ was seen at the *pond-mat*, *established mat* and *tundra* sites, with fluxes spanning zero at the *snout* and

*disturbed-mat* sites. $CH_4$ emission was highest at site *pond-mat* with some consumption measured at site *tundra*.

### 4.2 Halocarbon fluxes

The behaviour of the halocarbons over each surface type is broadly dictated by the compound type: mono-halogenated compounds ($CH_3Cl$, $CH_3Br$, $CH_3I$) were either consumed or fluctuated around zero, whereas polyhalomethanes ($CHBr_3$,

$CHCl_3$, $CH_2Br_2$) were emitted from all surfaces (Fig. 5). The mono-halogenated compounds were strongly and consistently drawn down at sites *established mat* and *tundra* with mean fluxes of -106 ±7 and -126 ±4 nmol m$^{-2}$ d$^{-1}$, respectively for $CH_3Cl$, -1.7 ±0.1 and -1.8 ±0.04 nmol m$^{-2}$ d$^{-1}$, respectively for $CH_3Br$ and -0.10 ±0.03 and -0.13 ±0.03 nmol m$^{-2}$ d$^{-1}$, respectively for $CH_3I$. A minor drawdown of $CH_3Cl$ (-11 ±5 nmol m$^{-2}$ d$^{-1}$) and $CH_3Br$ (-0.3 ±0.1 nmol m$^{-2}$ d$^{-1}$) occurred at site *pond-mat*, with near zero fluxes at site *snout*. Large variations in $CH_3I$ were recorded at sites *snout*, *pond-mat* and *disturbed mat*.

The polyhalomethanes were emitted from all surfaces, although the emission was relatively small at site *snout*. For $CHCl_3$, the site with the highest mean flux of 105 ±42 nmol m$^{-2}$ d$^{-1}$ was site *established mat*. However, due to the variation of $CHCl_3$ fluxes, this was not statistically different from the mean *tundra* flux of 74 ±33 nmol m$^{-2}$ d$^{-1}$ (p-value = 0.1). Fluxes of $CHBr_3$ were similarly varied, with the highest mean emission from site *disturbed-mat* of 0. 7 ± 0.3 nmol m$^{-2}$ d$^{-1}$ being statistically

similar to the flux at site *tundra* of 0.6 ± 0.1 nmol m$^{-2}$ d$^{-1}$ (p-value = 0.6). The highest mean flux of $CH_2Br_2$ was from site *tundra* (0.8 ±0.3 nmol m$^{-2}$ d$^{-1}$), with a smaller mean flux at sites *established mat*, *disturbed-mat* and *pond-mat* (all three had a



mean flux of 0.2 nmol m$^{-2}$ d$^{-1}$). CH$_2$BrCl was unquantified (Sect. 3.3) but was found to be emitted from all sites at similar relative magnitudes.

## 4.3 Relationships between halocarbon fluxes and physical, chemical and biological variables

To understand the different physical, chemical and biological factors associated with the halocarbon fluxes, correlations between them are presented in Fig. 6. Some of the chemical, physical and biological variables were strongly related to site location because the five sites differed in key factors such as vegetation cover and type. For example, OM, TN and TC contents were considerably higher at site *tundra* than the other sites (Fig. 4d-f). Some halocarbon fluxes also showed site-dependent variation such as the strong consumption of CH$_3$Cl and CH$_3$Br at site *established mat* and *tundra* compared to minor drawdown at the other sites. Because of the differences in physical variables and halocarbon fluxes at each site, we calculated correlation matrices for sites *disturbed-mat*, *established mat* and *tundra* separately (Fig. 6b-d). The difference between the correlations across all-sites (Fig. 6a) compared with the correlations at individual sites (Fig. 6b-d) showed that relationships between the different variables are not always consistent across sites.

### 4.3.1 Halocarbon intercorrelations

The two groups of halocarbons, the methyl halides (CH$_3$Cl, CH$_3$Br, CH$_3$I) and the polyhalomethanes (CHBr$_3$, CHCl$_3$, CH$_2$Br$_2$), show similar patterns of correlation (Fig. 6a). The methyl halides were all positively correlated with each other ($r > 0.62$, $p < 0.05$), as were the polyhalomethanes, but more weakly ($r > 0.54$; correlations with CHCl$_3$ were not significant, $p > 0.05$). All correlations between the two groups were negative (-0.18 $< r <$ -0.62; insignificant for CHBr$_3$ due to the weakness of the correlation; $0 > r >$ -0.2, $p > 0.05$). The negative correlation between the two groups indicated that, broadly, increased consumption of mono-halogenated compounds (i.e. more negative fluxes) correlated with increased production of poly-halogenated compounds.

The relationships within and between these two groups (methyl halides and polyhalomethanes) did not always persist across the three individual site analyses. For example, at site *disturbed mat*, all the halocarbons except CH$_3$I were positively correlated (Fig. 6b) suggesting higher emission of the polyhalomethanes occurred with lower consumption of CH$_3$Cl and CH$_3$Br, contrary to the all-site relationship. Furthermore, there were instances where correlations across all sites appeared to be driven by the large size of their relationship at one site. For example, the weak positive correlation across all sites between the haloforms (CHX$_3$; X = Cl, Br), CHBr$_3$ and CHCl$_3$ ($r = 0.29$), was inflated by their strong positive correlation at site *disturbed mat* ($r = 0.98$) which masked their negative correlation at sites *established mat* and *tundra* ($r =$ -0.29 and -0.57, respectively). The results from the individual site analyses demonstrate the importance of investigating differences in halocarbon patterns by small scale geography.



### 4.3.2 Correlations of methyl halides and chemical, physical and biological variables

Across all sites, the mono-halogenated compounds were negatively correlated with OM, TC, TN and bacterial cell numbers with the strongest correlation for $CH_3Cl$ ($r < -0.60$), and weakest for $CH_3I$ ($r < -0.39$), indicating greater methyl halide

consumption (i.e. more negative fluxes) occurred with higher concentrations of OM, TC, TN and bacterial cells in the sediment/ soil. This was largely driven by high methyl halide consumption at sites *established mat* and *tundra* where OM, TC, TN and bacterial cell contents were highest. The relationship broadly persisted at site *established mat* (Fig. 6c), but not at sites *disturbed mat* and *tundra* (Fig. 6b, d). Across all sites, the methyl halides were negatively correlated with water content and water table depth ($r < -0.45$; $CH_3I$ and water table depth are insignificant) showing higher methyl halide consumption (i.e. lower fluxes)

where water contents were higher, but the water table deeper. $CH_3Cl$ and $CH_3Br$ were negatively correlated with $CO_2$ ($r = -0.41$ and $-0.45$, respectively) indicating increased consumption correlated with $CO_2$ fluxes tending from consumption to production (i.e. becoming more positive). The opposite relationship was seen with $CH_4$ ($r = 0.43$ and $0.37$), broadly indicating increased $CH_3Cl$ and $CH_3Br$ consumption occurred with smaller $CH_4$ fluxes, i.e. tending towards consumption.

### 4.3.3 Correlations of polyhalomethanes and chemical, physical and biological variables

Compared to the methyl halides, the polyhalomethanes ($CHCl_3$, $CHBr_3$ and $CH_2Br_2$) generally showed opposite and weaker correlations with positive correlations with OM, TC, TN contents, bacterial cell numbers and water content (Fig. 6a). However, many of the correlations were not significant for the three gases. Across all sites, $CHCl_3$ and $CHBr_3$ were not strongly or significantly correlated with any variable ($-0.4 < r < 0.4$, $p > 0.05$) except bacterial cell numbers with $CHCl_3$ ($r = 0.67$) and

TC content ($r = 0.41$) and water content ($r = 0.56$) with $CHBr_3$. However $CH_2Br_2$ was strongly positively correlated with water, OM, TC and TN contents ($r > 0.7$), showing that increased emission of $CH_2Br_2$ was correlated with increased OM, TC, TN and water contents. $CH_2Br_2$ was negatively correlated with $CH_4$ contents ($r = -0.41$) indicating greater $CH_2Br_2$ emission when $CH_4$ fluxes tended towards consumption (i.e. lower fluxes). Similarly to the methyl halide compounds, some of the all-site relationships for the polyhalomethanes were also present within an individual site and others were not (Fig. 6 b-d). For

example, an interesting intra-site trend at site *disturbed mat* is the very strong positive correlation between the three polyhalomethanes and temperature and OM content ($r > 0.9$).

### 5 Discussion

### 5.1 Influence of exposure age on halocarbon fluxes from the proglacial environment

Terrestrial halocarbon fluxes are predominantly driven by biological processes (e.g. Amachi et al., 2001; Dimmer et al., 2001;

Redeker and Kalin, 2012), and a lower prevalence of abiogenic processes which often involve oxidation of organic matter



(Huber et al., 2009; Keppler et al., 2000). Both of these processes would suggest that increasing soil development would be an important driver of halocarbon fluxes. As such, immature soils, such as those exposed by retreating ice, may be assumed to have minor trace gas fluxes in comparison to more developed soils with established biota. Further, one might expect an increase in flux magnitude as the soil develops with increasing exposure age, i.e. with greater distance from the glacier terminus. Our

study does indicate that some soil development is required for most halocarbon fluxes, with the lowest mean fluxes of all gases (except for $CH_3I$) measured at the youngest site (site *snout*; ~ 5 years), which has no vegetation and very little organic matter (0.1 % of soil). Whereas, site *tundra*, the oldest site (approximately 1950 years exposure), with full coverage of vegetation, high bacterial cell numbers (3.2 x $10^8$ cells [g sediment]$^{-1}$), and more soil development (e.g. 6.0 % OM content) had the highest mean consumption of $CH_3Cl$, $CH_3Br$ and $CH_3I$ and the highest mean emission of $CH_2Br_2$. However, there were exceptions to

this trend which imply that soil development is not the only driver of halocarbon fluxes. For example, consumption rates of $CH_3Cl$, $CH_3Br$ and $CH_3I$ at site *established mat* were similar to those seen at site *tundra*, despite the large difference in soil development (TC, TN and OM contents; Fig. 4). Further, fluxes at sites *established mat* and *tundra* of $CH_3Cl$ (-106 ± 7 and -126 ± 4 nmol m$^{-2}$ d$^{-1}$, respectively) and $CH_3Br$ (-1.7 ± 0.1 and -1.8 ± 0.04 nmol m$^{-2}$ d$^{-1}$, respectively) were within the range measured at a well-established coastal tundra site in Alaska where flooded and drained sites had respective mean fluxes of -

14 to -620 nmol m$^{-2}$ d$^{-1}$ for $CH_3Cl$ and +1.1 to -9.8 nmol m$^{-2}$ d$^{-1}$ for $CH_3Br$ (Rhew et al., 2007). However, fluxes of $CH_3I$ at site *tundra* and *established mat* (-0.13 ± 0.03 and -0.10 ± 0.03 nmol m$^{-2}$ d$^{-1}$) were negative, contrasting a mean emission of 4.0 nmol m$^{-2}$ d$^{-1}$ measured from Alaskan tundra (Rhew et al., 2007; Fig. 7).

This pattern whereby sites with younger, less-developed soils have similar fluxes to the older and developed soil of site *tundra*

also occurred for $CHCl_3$ and $CHBr_3$ where the highest mean fluxes were measured at site *established mat* and site *disturbed mat*, respectively, but were statistically similar to the flux measured at site *tundra* (p = 0.1, 0.6, respectively). This is even more surprising for site *disturbed mat* which is completely bare of vegetation and has comparatively low bacterial cell numbers (Fig. 4g). Terrestrial fluxes of $CHBr_3$ have rarely been measured (see Sect. 5.2), whereas $CHCl_3$ emissions have been recorded, including from the Alaskan tundra where the average flux was 45 nmol m$^{-2}$ d$^{-1}$ (Rhew et al., 2008). Mean emissions of $CHCl_3$

were larger at sites *tundra* and *established mat* and similar at site *disturbed mat* (74, 106 and 43 nmol m$^{-2}$ d$^{-1}$, respectively). Considerable variability of $CHCl_3$ fluxes were measured, with the range for site *tundra* of 23 to 128 nmol m$^{-2}$ d$^{-1}$ and the range for site *established mat* of 64 to 183 nmol m$^{-2}$ d$^{-1}$. This variability in $CHCl_3$ fluxes is less than, but comparable to, the variation measured at the Alaskan tundra of <1 to 260 nmol m$^{-2}$ d$^{-1}$ (Rhew et al., 2008). We have demonstrated that younger surfaces can be sources of $CHCl_3$ and $CHBr_3$ and sinks of $CH_3Cl$ and $CH_3Br$ despite their lesser soil development and lower microbial

and plant presence. In particular, it appears the presence of cyanobacterial mats negates the requirement for a more developed soil. To our knowledge, no studies have been conducted upon halogenated trace gas fluxes from cyanobacteria mats or freshwater cyanobacteria, although marine cyanobacteria have been suggested to be involved in production of $CH_2Br_2$, $CHBr_3$ and $CH_3I$ (Karlsson et al., 2008; Roy et al., 2011). Determining if cyanobacteria themselves, or other microorganisms present in the mat are responsible for the elevated fluxes was beyond the scope of this study.



### 5.2 Terrestrial emission of typically marine-origin brominated compounds

A second novel finding of this study was the emission of CHBr₃ and CH₂Br₂ across the glacial forefield, with very small emissions at site *snout*, but more appreciable fluxes at all other sites (Fig. 5e-f). CHBr₃ and CH₂Br₂ are typically attributed to marine sources (Law et al., 2006). However, there have been limited observations of emission of both compounds from terrestrial environments. CHBr₃ has been observed to be emitted from rice paddies, with algae in the water column as the suggested source, however a rice-mediated production mechanism was not discounted (Redeker et al., 2003). CH₂Br₂ emissions have been observed from wet temperate peatlands, with no production mechanism suggested (Dimmer et al., 2001). Emission of CHBr₃ has been observed, but not quantified, from the transitional terrestrial-marine environment of a coastal wetland, where it was shown to be abiogenic in origin (Wang et al., 2016). Further, abiogenic production of CHBr₃ through the oxidation of organic matter by Fe(III) and H₂O₂ when halide ions are present has been documented in a laboratory based soil study (Huber et al., 2009). The largest flux of CH₂Br₂ is measured at site *tundra* which is analogous to an Arctic peatland ecosystem, and thus complements the emissions measured from temperate peatlands in Ireland (Dimmer et al., 2001). Our results provide further evidence of the emission of these two compounds in a terrestrial environment, and the first evidence of terrestrial emission of these compounds in the Arctic.

### 5.3 Controls on halocarbon fluxes in the proglacial environment

#### 5.3.1 Biological consumption of methyl halides and abiogenic production of CH₃I

Methyl halides were primarily consumed on the glacier forefield, with all three compounds consistently consumed at sites *established mat* and *tundra* but with fluxes of CH₃I in both directions at sites *snout*, *pond-mat* and *disturbed mat*. The strong inter-correlations between different methyl halides suggest a similar consumption mechanism, particularly between CH₃Cl and CH₃Br. Strong correlations between CH₃Cl and CH₃Br have been found elsewhere, including in the Alaskan tundra, with similar suggestions of common consumption mechanisms or common limiting factors (Rhew et al., 2007). We suggest that the consumption of all three methyl halides observed across the forefield is driven by prokaryotic degradation, which is supported by methyl halide fluxes being correlated with bacterial cell concentrations ($r < -0.52$) and net microbial respiration (CO₂ emission; $r < -0.41$, not significant for CH₃I). Both biogenic and abiogenic (through organic matter oxidation) soil production mechanisms of CH₃I have previously been demonstrated (Amachi et al., 2001; Keppler et al., 2000). However, these mechanisms are not strongly supported here as CH₃I is emitted at the sites (*snout*, *pond-mat*, *disturbed mat*) with the lowest bacterial concentrations and lowest organic matter contents (0.1-0.6 %). Identifying the CH₃I production mechanism would require further study.

#### 5.3.2 Inconclusive influence of water content on methyl halide fluxes

Several studies have identified the importance of soil water content for CH₃X fluxes, with very low water contents limiting biological activity and high water contents limiting the mass transfer of reactants during CH₃X formation and degradation





(Khan et al., 2012; Rhew et al., 2010; Teh et al., 2009). We find that increasing water content was correlated to greater consumption of $CH_3X$ across all sites, despite high water contents (> 40 % v/v). This is driven largely by high water contents at site *tundra* where the highest consumption of $CH_3X$ was found, presumably due to the more developed soils and biota at this site. Within site *tundra* the relationship with water content persists, in contrast to the Alaskan tundra studies which found

that decreasing water content was the key factor causing increased consumption of $CH_3Cl$ and $CH_3Br$ (Teh et al., 2009). Our results are not consistent with this finding perhaps due to the noise caused by a small within-site sample size (n = 8) coupled with a smaller range of water volumes measured here (40-60 %, compared to < 30 to > 70 % in the Alaskan study). Further, the relationship between $CH_3X$ and water content implied greater consumption in more anaerobic soils, however, higher consumption of $CH_3X$ was found to occur where fluxes of $CH_4$ are tending towards the aerobic process of consumption, as

found in the Alaskan tundra (Rhew et al., 2007). The contradiction between water content and aerobic $CH_4$ consumption shown here further indicates that more within-site data is required, as the disparity in the $CH_3X$ fluxes of the different sites drives the all-site relationships.

### 5.3.3 Biogenic and abiogenic production of polyhalogenated species

Biogenic production mechanisms of $CHCl_3$, $CHBr_3$ and $CH_2Br_2$ are shared (haloperoxidase activity), as is the abiogenic

production mechanism of the haloforms ($CHX_3$; Huber et al., 2009; Manley, 2002). If either biogenic or abiogenic processes were the sole source of the polyhalogenated species, then we would expect that, at least, $CHX_3$ fluxes would be correlated. However, $CHCl_3$ and $CHBr_3$ are not well correlated across all sites (*r* = 0.29, p > 0.05) suggesting different sources of these compounds within or between sites. Here, $CHCl_3$ is strongly correlated to bacterial cell numbers, but $CHBr_3$ is not, which tentatively suggests that $CHCl_3$ is produced biologically. At sites *established mat* and *tundra*, $CHCl_3$ and $CHBr_3$ were not

significantly correlated suggesting multiple sources or a possible unknown consumption process. There is no evidence prior to this study that terrestrial or freshwater cyanobacteria are involved in halocarbon production. However, marine cyanobacteria have been implicated in the production of $CHBr_3$ and $CH_2Br_2$ and the bromoperoxidase enzyme has been identified in some marine species (Johnson et al., 2011; Karlsson et al., 2008). The highest emissions of $CH_2Br_2$ at site *tundra*, could be due to a different microbial community make-up or a plant-mediated process. We suggest that a possible mixture of abiogenic and

biogenic production mechanisms are responsible for $CHCl_3$ and $CHBr_3$ emissions, whereas $CH_2Br_2$ emissions seem more likely to be driven biologically.

### 5.4 Glacial forefields as a source and sink of halocarbons?

Determining the local or regional importance of the proglacial halocarbon fluxes would require further study into diurnal,

seasonal and spatial variations. However, estimations of the yearly regional source or sink of each gas is still worthwhile, particularly for $CHBr_3$ and $CH_2Br_2$ for which no prior fluxes have been measured from terrestrial Arctic environments. We calculate an Arctic tundra source of 0.11 and 0.09 Gg Br yr$^{-1}$ for $CHBr_3$ and $CH_2Br_2$, assuming that no production occurs



outside of the growing season (Sect. 3.6.2). The sources are minor compared to the estimated global sources of 120-820 Gg Br yr$^{-1}$ for CHBr$_3$ and 57-100 Gg Br yr$^{-1}$ for CH$_2$Br$_2$, which are primarily oceanic in origin (Carpenter et al., 2014; Engel et al., 2018). To determine if our tundra source has regional significance, we estimate the proportion of the global flux that may occur in the Arctic assuming the global source is equally distributed over the earth's surface, and using an Arctic surface area

(area north of the Arctic circle) of 4% of the earth's total. The tundra source would be an estimated 0.3-2 % of CHBr$_3$ and 2-4 % of CH$_2$Br$_2$ of the estimated total Arctic source of 5-33 Gg Br yr$^{-1}$ for CHBr$_3$ and 2-4 Gg Br yr$^{-1}$ for CH$_2$Br$_2$. Further, global sources are dominantly marine, and although Arctic macroalgae are a source of both gases (Laturnus, 1996), polar oceans as a whole have been suggested to be a sink (e.g. Chuck et al., 2005; Ziska et al., 2013; Fig. 7). Therefore, a terrestrial Arctic source could be more regionally important than estimated here.

For the other halocarbons analysed across the glacial forefield, we calculated a potential proglacial regional flux from an estimated Arctic proglacial land area (Sect. 3.6.1). Small net sinks of 8 tonnes of CH$_3$Cl, 0.2 tonnes of CH$_3$Br, 0.01 tonnes of CH$_3$I, and small net sources of 18 tonnes of CHCl$_3$, 0.2 tonnes of CH$_2$Br$_2$ and 0.3 tonnes of CHBr$_3$ were calculated. All of these are minor compared to global fluxes, due to the relatively small area of land covered by proglacial surfaces. Fluxes from

the Alaskan tundra, which were similar to our fluxes for *established mat* and *tundra*, were found to be regionally important, where they represent the equivalent of approximately 20-25 % and 10-15 % of the seasonal variation in the Arctic troposphere of CH$_3$Cl and CH$_3$Br, respectively (Rhew et al., 2007). However, our estimated proglacial land surface area is 2 orders of magnitude smaller than the estimated area of Arctic tundra (7.3 x 10$^{12}$ m$^2$) meaning even within the Arctic troposphere the proglacial sink of CH$_3$Cl and CH$_3$Br is insignificant.

Although our daily fluxes are likely an underestimate (due to calculation from concentration change over 1 hour, after the rate of change had slowed; Sect. 3.3), the magnitude of this underestimate will not be large enough to alter the significance of the total gas fluxes regionally. Despite all halocarbons studied here appearing to represent only minor fluxes globally and regionally, this study has shown the potential for younger surfaces to be involved in halocarbon flux processes which may

become more important due to expansion of these surfaces under future warming.

## 6 Conclusions

We present the first measurements of halocarbon fluxes from proglacial land surfaces, showing an overall net sink of CH$_3$Cl, CH$_3$Br and CH$_3$I and net source of CHCl$_3$, CHBr$_3$ and CH$_2$Br$_2$. Relatively young, under-developed soils exposed by glacial retreat can have similar fluxes of halocarbons to older, more developed soils, particularly where cyanobacterial mats have

formed. We have shown that surfaces covered in these mats are sinks of CH$_3$Cl, CH$_3$Br, CH$_3$I and sources of CHCl$_3$ and CHBr$_3$. The latter two gases also show appreciable fluxes even from bare sediment adjacent to cyanobacterial mats. This is the first research known to us conducted on terrestrial cyanobacteria, and additionally we have provided comparatively rare

terrestrial flux measurements of $CHBr_3$ and $CH_2Br_2$. Future work should: identify if cyanobacteria themselves or other microbes are responsible for the high fluxes over the mats; improve the spatial and temporal distribution of these measurements, including conducting measurements outside the growing season; conduct gas analyses at less than 1 hour intervals to reduce the suspected underestimation of the flux calculations; and identify if other terrestrial environments emit

$CHBr_3$ and $CH_2Br_2$, particularly in areas where the fluxes might be higher (i.e. in more developed and more active soils) and therefore more regionally important. The significance of proglacial fluxes may become more important in the future with continuing change in the Arctic and the resultant retreat of glacial systems and exposure of proglacial land.

## 7 Data availability

Raw data supporting the conclusions and used to create the figures of this manuscript are available at
doi: 10.6084/m9.figshare.8081129

## 8 Author contributions

MLM and JLW conceived the study. MLM, DY and GLG conducted the field work. SO, CL and OH supplied resources integral to the fieldwork. MLM conducted the analysis with assistance from DY. MLM prepared the manuscript with review and comments provided by all authors.

## 9 Competing Interests

The authors declare that they have no conflict of interest.

## 10 Acknowledgements

We would like to thank Nathan Chrismas for assistance in the field, and Nicholas Cox and Malcolm Airey for logistical support at the UK Arctic Research Station in Svalbard. Dr. Fotis Sgouridis is also thanked for laboratory support in the LOWTEX
Laboratory at the University of Bristol. This work was funded by a GW4+ Doctoral Training Partnership studentship awarded to MLM by the Natural Environment Research Council [NE/L002434/1], with additional funding from an SSF Arctic Field Grant awarded to MLM and CL [257104/E10].

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

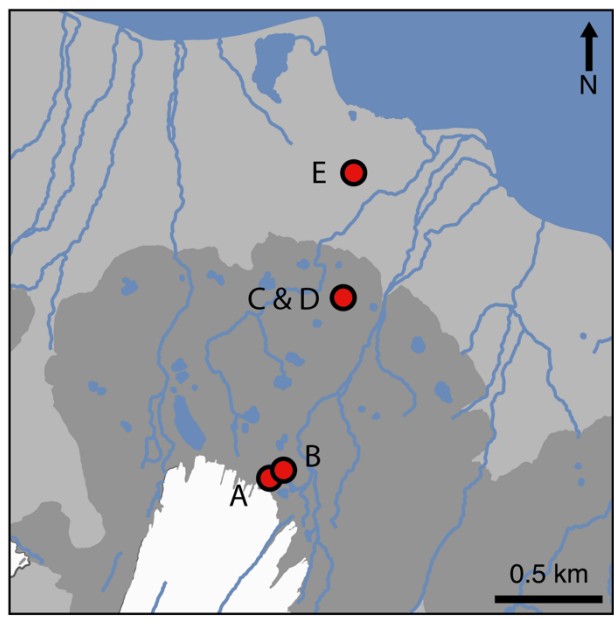

**Figure 1: Locations of sites *snout* (A), *pond mat* (B), *disturbed mat* (C), *established mat* (D) and *tundra* (E) on the proglacial forefield of Midtre Lovenbreen glacier (white). The moraine field is denoted in dark grey, the maximum extent of which marks the furthest extent of the glacier during the Little Ice Age. Data used to create the base map from: Norwegian Polar Institute (2014).**



**Figure 2: The visible differences in land-surface type and colonising species at site *snout* (a), *pond-mat* (b), *disturbed mat* (c), *mat* (d) and *tundra* (e), and a schematic diagram of the flux chamber's design showing sampling ports, fan and temperature logger (f). The width of the chamber collar in (a)-(e) is 0.39 m.**



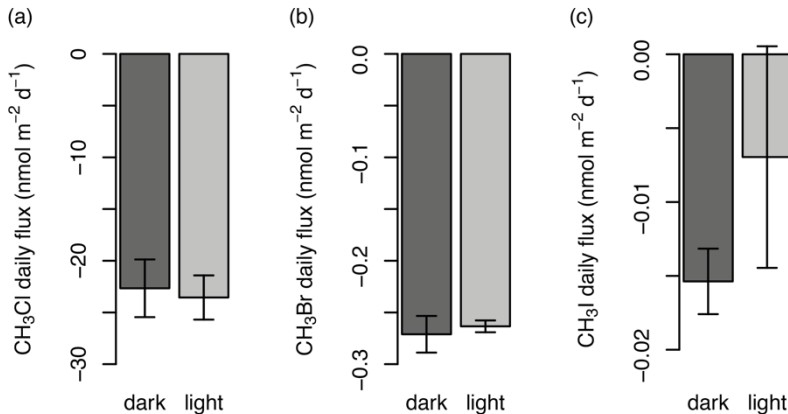

**Figure 3: Comparison of the flux in nmol m⁻² d⁻¹ of gas in un-darkened (light) and darkened (dark) chambers for CH3Cl (a), CH3Br (b), CH3I (c) from preliminary experiments in 2016.**

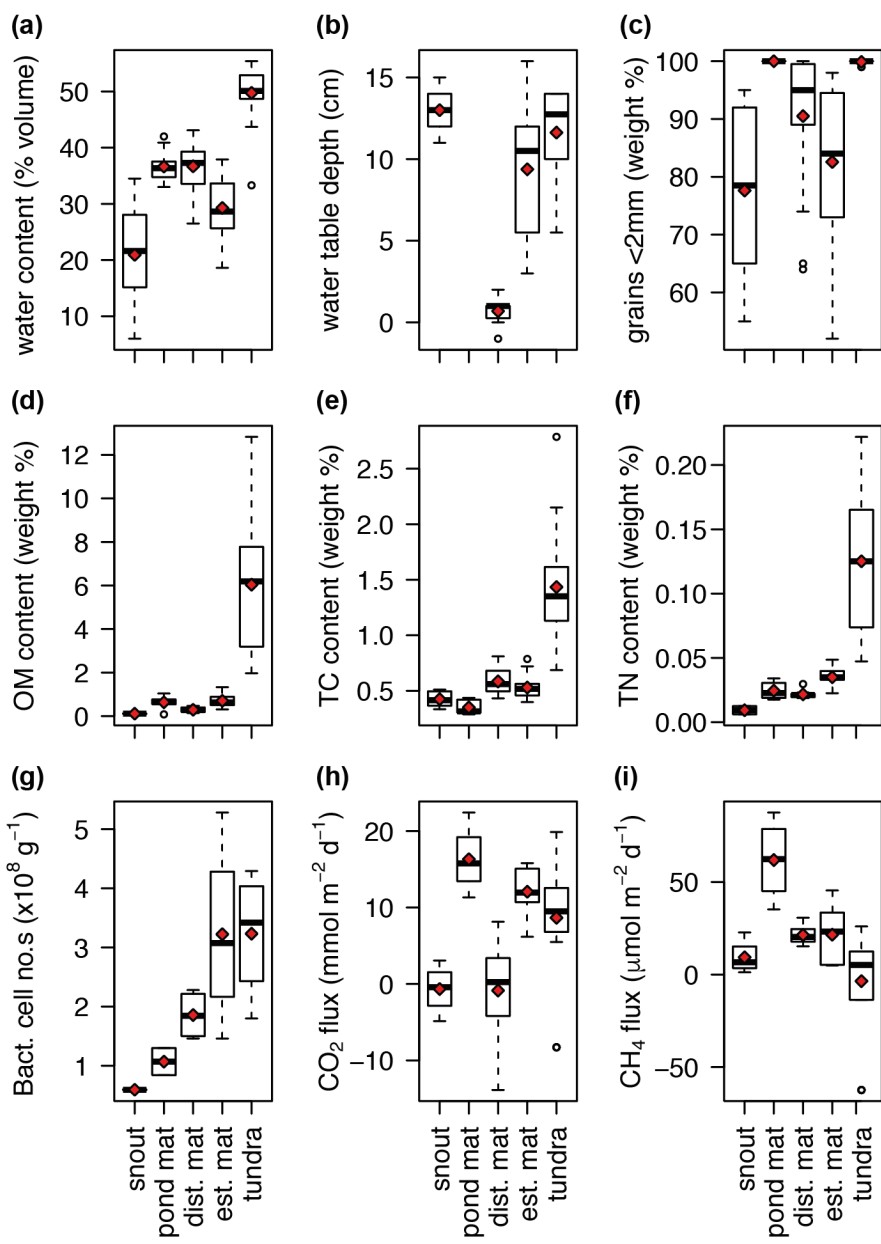

**Figure 4: Variation at each site of soil water content (a), water table depth (b), weight % of grains < 2 mm diameter (c), organic matter content (d), total carbon content (e), total nitrogen content (f), bacterial cell numbers (g), $CO_2$ flux (h) and $CH_4$ flux (i). Horizontal black bar represents the median, red diamonds the mean for each site, open circles are outliers. "dist. mat" is site** *disturbed mat***, "est. mat" is site** *established mat***. Water table was not measurable for site** *pond-mat* **due to rocky ground.**





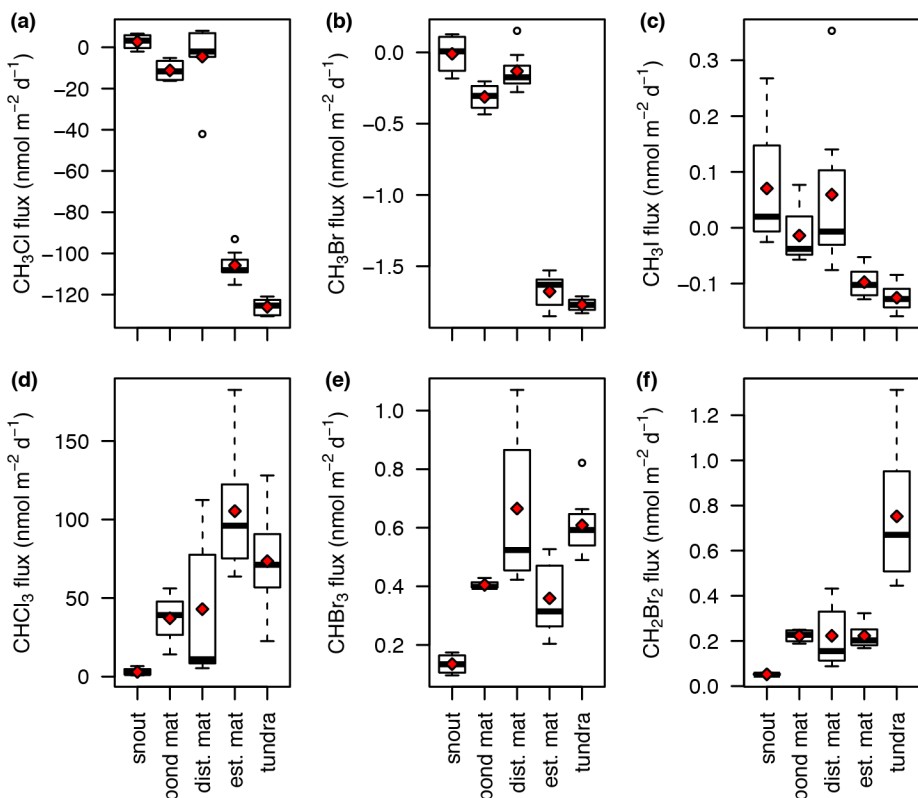

**Figure 5: Daily fluxes (nmol m$^{-2}$ d$^{-1}$) at each site of CH$_3$Cl (a), CH$_3$Br (b), CH$_3$I (c), CHCl$_3$ (d), CHBr$_3$ (e), CH$_2$Br$_2$ (f). Red diamonds represent the mean flux for each site. "dist. mat" is site *disturbed mat*, "est. mat" is site *established mat*.**





**Figure 6: Correlations between halocarbon fluxes and the chemical, physical and biological variables across all sites (a), site *disturbed mat* (b), site *established mat* (c) and site *tundra* (d). White stars indicate correlations with 95 % confidence (p < 0.05).**


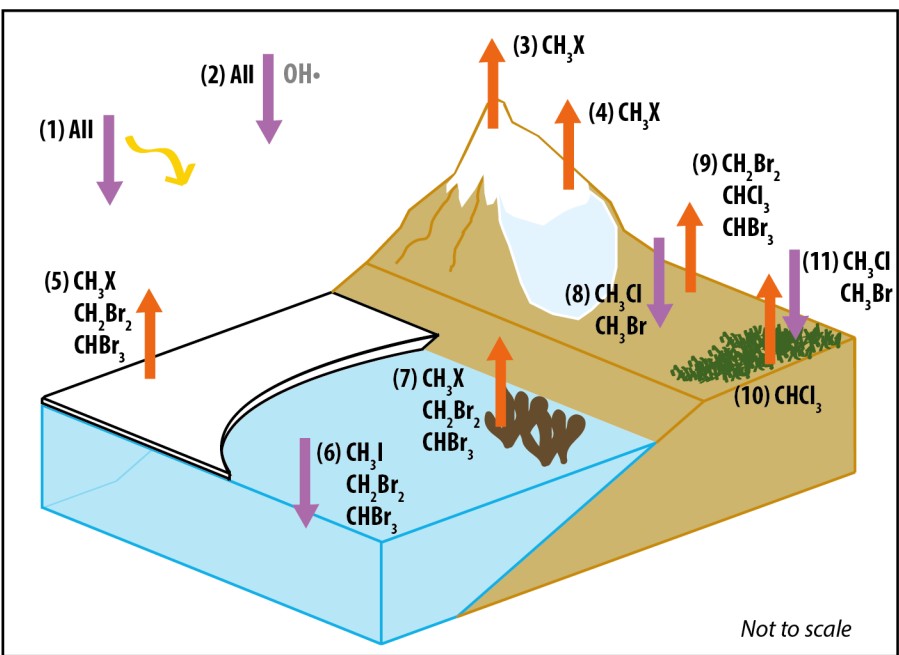

**Figure 7: Schematic diagram summarising natural sources and sinks for the 6 halocarbons of interest in polar regions. The sources/ sinks are as follows: (1) UV photolysis, (2) reaction with OH•, (3) photochemistry in snow, (4) microbial activity in snow, (5) sea-ice microalgae, (6) open ocean, (7) macroalgae, (8) proglacial sink, (9) proglacial source, (10) tundra source, (11) tundra sink. ($CH_3X$ = $CH_3Cl$, $CH_3Br$, $CH_3I$). References for the presence of each flux is as follows: 1, 2 (see Montzka et al. (2011) for review), 3 (Swanson et al., 2007), 4 (Redeker et al., 2017), 5 (Laturnus et al., 1998; Sturges et al., 1993), 6 (Stemmler et al., 2014; Ziska et al., 2013), 7 (Laturnus, 1996, 2001), 8 and 9 (this study), 10 (Albers et al., 2017; Rhew et al., 2008), 11 (Rhew et al., 2007).**





**Table 1: The standard concentration, limit of quantification (variance) and limit of detection (LOD) for each gas analysed, with the target ion and qualifier ion(s) shown for gases analysed by GCMS. (\*) ppt for the halocarbons, ppm for $CO_2$ and $CH_4$. (NA) not applicable to the method of measurement. "equi." is short for equivalent.**

|  | Units | $CH_3Cl$ | $CH_3Br$ | $CH_3I$ | $CHCl_3$ | $CHBr_3$ | $CH_2Br_2$ | $CO_2$ | $CH_4$ |
|---|---|---|---|---|---|---|---|---|---|
| **Target ion** | m/z | 52 | 94 | 142 | 83 | 171 | 174 | NA | NA |
| **Qualifier ion(s)** | m/z | 50 | 96 | 127 | 85 | 173, 175 | 93, 95 | NA | NA |
| **Standard conc.** | ppt / ppm* | 530 | 6.4 | 0.47 | 16.7 | 2.8 | 1.3 | 405.6 ±5% | 194.7 ±5% |
| **Variance (n=49)** | % | 2 | 1 | 3 | 1 | 3 | 2 | 1.7 | 1.1 |
| **Variance equi.** | nmol m$^{-2}$ | 0.1 | 0.0007 | 0.0001 | 0.002 | 0.0008 | 0.0002 | 0.02 | 0.2 |
|  | nmol m$^{-2}$ d$^{-1}$ | 2 | 0.02 | 0.003 | 0.05 | 0.02 | 0.005 | 0.5 | 4.7 |
| **LOD** | ppt | 1.4 | 0.3 | 0.01 | 0.18 | 0.38 | 0.08 | 0.32 | 0.16 |
| **LOD, equi.** | nmol m$^{-2}$ | 0.01 | 0.003 | 0.0001 | 0.002 | 0.004 | 0.0009 | 0.003 | 0.002 |
| **LOD, equi.** | nmol m$^{-2}$ d$^{-1}$ | 0.3 | 0.07 | 0.003 | 0.04 | 0.09 | 0.02 | 0.07 | 0.05 |

