# Peer review of "Consumption of CH3Cl, CH3Br and CH3I and emission of CHCl3, CHBr3 and CH2Br2 from the forefield of a retreating Arctic glacier"

_Atmospheric Chemistry and Physics, 2019_

## Referee Comment (RC1) · Anonymous Referee #1 · 11 Jan 2020

This is a nicely written, well organized paper in which the fluxes of methyl halides and polybrominated methanes are determined from land surfaces exposed by glacial retreat in the Arctic. Multiple sites with different ages since becoming uncovered by an overburden glacier are studied to determine the influence of soil ecosystem development on trace gas fluxes. A number of interesting conclusion are derived that are novel and will be informative for ACP readers. I recommend publishing after further consideration of some minor issues.

I find the abstract not as informative as it should be. The quantitative information is provided without context and without mention of implications other than the one that could

have been made even before the work was performed: "With future glacial retreat and the expansion of these surfaces, these fluxes may become important in the future". Furthermore, the quantitative results provided by the paper actually suggest that these fluxes are quite small and not that different from non-proglacial land. Hence, it is very difficult to imagine that they will become significant on a broader scale at some point in the future even with a large increase in pro-glacial land surface area. The discussion in section 5.4 seems much more straightforward in describing the implications of these fluxes. Recommendation: include many of the very informative qualitative conclusions you mention throughout the discussion section that are the result of this work (many paragraphs in section 4 and 5 starts or ends with one of these nuggets). Do not overstate the potential future importance of these fluxes; what might be viewed as a negative result here is still very useful and informative. Finally, if quantitative results remain in the abstract, mention also for context the magnitude of similar fluxes in other regions of the Arctic for context.

Details: Define the terms proglacial and forefield for this audience.

Radiocarbon dating at the tundra site indicated a date of exposure of 1850-1926 BP (before present?), so it is not clear where the "approximately 1950 year old" age comes from (abstract and elsewhere).

Line 4 of intro: this statement is not true for CH3Cl and CH3Br until you describe them as "the most important *natural* sources of chlorine and bromine to the troposphere".

It is not explicitly clear if the ballast synthetic air which was drawn from to maintain pressure in the chambers during sampling was the "zero air" mentioned earlier, and if this air was de-humidified and CO2-free? I wonder if some inconsistent changes in fluxes during the 2-hr experiments might have been caused by changes in CO2 concentrations and humidity in the chamber.

Consider in Figure 7 highlighting somehow the fluxes discussed in this paper.

---

## Referee Comment (RC2) · Anonymous Referee #2 · 21 Feb 2020

This is a very well written and documented manuscript that describes the consumption/emission of various organic short-lived halocarbons, CO2, and CH4 in a variety of environments within a retreating glacier forefield. The results suggest these polar glacial regions are not a strong source or sink of these gases, however it lays the ground work for future studies to explore this in other glacial areas and to further examine the role of terrestrial cyanobacterial production of halogenated trace gases. I recommend publishing this manuscript after the authors address the minor points below.

The flux values in the abstract need some context, e.g., how significant are they in

[Figure]

terms of sources or sinks or how do they compare with other measurements if available.

Were there any lab tests of potential impacts on storing the air samples in the vials or bags prior to analysis?

On p. 4, line 19 the radiocarbon age is 1850-1926 for the tundra site, but the abstract says approximately 1950 year old tundra.
* * *

---

## Author Comment (AC1) · 17 Apr 2020

**Response to reviewers**

We would like to thank both anonymous reviewers and the editor for their helpful comments which have improved the quality of this manuscript.

**Responses to anonymous referee #1**

**(1) I find the abstract not as informative as it should be. The quantitative information is provided without context and without mention of implications. Furthermore, the quantitative results provided by the paper actually suggest that these fluxes are quite small and not that different from non-proglacial land. Hence, it is very difficult to imagine that they will become significant on a broader scale at some point in the future even with a large increase in pro-glacial land surface area. Recommendation: include many of the very informative qualitative conclusions you mention throughout the discussion section that are the result of this work (many paragraphs in section 4 and 5 starts or ends with one of these nuggets). Do not overstate the potential future importance of these fluxes; what might be viewed as a negative result here is still very useful and informative. Finally, if quantitative results remain in the abstract, mention also for context the magnitude of similar fluxes in other regions of the Arctic for context.**

Thank you for these constructive comments. In light of your suggestions, and with a related comment from reviewer 2, we have amended the abstract to highlight some of the other conclusions in this manuscript rather than the fluxes which have been removed from the abstract. The changes to the abstract are as follows (page 1, lines 22-30):

"Bromoform ($CHBr_3$) and dibromomethane ($CH_2Br_2$) have rarely been measured from terrestrial sources but were here found to be emitted across the forefield. Novel measurements conducted on terrestrial cyanobacterial mats covering relatively young surfaces showed similar measured fluxes to the oldest, vegetated tundra sites for $CH_3Cl$, $CH_3Br$ and $CH_3I$ (which were consumed) and for $CHCl_3$ and $CHBr_3$ (which were emitted). Consumption rates of $CH_3Cl$ and $CH_3Br$ and emission rates of $CHCl_3$ from tundra and cyanobacterial mat sites were within the ranges reported from older and more established Arctic tundra elsewhere. Rough calculations showed total emissions and consumptions of these gases across the Arctic were small relative to other sources and sinks due to the small surface area represented by glacier forefields. We have demonstrated that glacier forefields can consume and emit halocarbons despite their young age and low soil development, particularly when cyanobacterial mats are present."

**(2) Define the terms proglacial and forefield for this audience.**

Thank you for pointing this out. For simplicity we have decided to just use one of these terms ("glacier forefield" and sometimes more simply "forefield") and so have made changes throughout the manuscript to remove the "proglacial" term. "Forefield" is now defined where it first appears in the manuscript on page 1, line 18-19:

"…we measured halocarbon fluxes across the glacier forefield (the area between the present day position of a glacier's ice-front and that at the last glacial maximum)…"

**(3) Radiocarbon dating at the tundra site indicated a date of exposure of 1850-1926 BP (before present?), so it is not clear where the "approximately 1950 year old" age comes from (abstract and elsewhere).**

The tundra age of "approximately 1950 years old" was calculated as the age from the present day (specifically, the date of the fieldwork, 2017), i.e. the radiocarbon age, which is dated from 1 January 1950, plus the difference between 1950 and 2017. This was to keep the approximate ages of all land surfaces consistent with each other, as some of the surfaces are younger than the year 1950 and therefore have ages estimated from the year fieldwork was conducted. The age from today (rather than from BP) of the tundra would be 1917-1993 years old, which we approximated at 1950 years old. This has been clarified in the text on page 4, line 19-20:

"Radiocarbon dating near site *tundra* (~70 m west) has provided a date of exposure of 1850-1926 BP (Before Present, defined as 1$^{st}$ January 1950 by the radiocarbon age scale; Hodkinson et al., 2003). This is equivalent to 1917-1993 years older (or approximately 1950 years) than the year of analysis (2017)."

**(4) Line 4 of intro: this statement is not true for CH3Cl and CH3Br until you describe them as "the most important \*natural\* sources of chlorine and bromine to the troposphere".**

Thank you for pointing this out, the description has now been corrected in the manuscript as follows (page 2, line 4):

"Methyl chloride ($CH_3Cl$) and methyl bromide ($CH_3Br$) are the most important natural sources of chlorine (16%) and bromine (50%) to the troposphere and are important contributors to stratospheric ozone loss (Carpenter et al., 2014)."

**(5) It is not explicitly clear if the ballast synthetic air which was drawn from to maintain pressure in the chambers during sampling was the "zero air" mentioned earlier, and if this air was de-humidified and CO2-free? I wonder if some inconsistent changes in fluxes during the 2-hr experiments might have been caused by changes in CO2 concentrations and humidity in the chamber.**

The synthetic air used was grade 5.0 so it was de-humidified and $CO_2$ free (with a purity of 99.999%). This has been clarified in the manuscript where the zero air is first mentioned in the manuscript (page 6, line 8-9):

"All sample bags were flushed three times with Grade 5.0 synthetic zero air (dry and $CO_2$-free) prior to use, with laboratory testing indicating this removed any background contamination."

**(6) Consider in Figure 7 highlighting somehow the fluxes discussed in this paper.**

Thank you for this suggestion. The proglacial fluxes analysed in this paper have been highlighted in the manuscript figure on page 29.
"

[Figure]

Figure 7: Schematic diagram summarising natural sources and sinks for the 6 halocarbons of interest in polar regions with fluxes measured in this manuscript (8 and 9) highlighted in orange."

**Responses to anonymous referee #2**

**(1) The flux values in the abstract need some context, e.g., how significant are they in terms of sources or sinks or how do they compare with other measurements if available.**

We thank the reviewer for this comment. Following similar comments from the other reviewer, we decided to re-draft the abstract without focussing on actual flux values. We have also included a comment on how the measured fluxes relate to other sources and sinks in the Arctic. The changed text is now as follows (page 1, lines 22-30):

"Bromoform ($CHBr_3$) and dibromomethane ($CH_2Br_2$) have rarely been measured from terrestrial sources but were here found to be emitted across the forefield. Novel measurements conducted on terrestrial cyanobacterial mats covering relatively young surfaces showed similar measured fluxes to the oldest, vegetated tundra sites for $CH_3Cl$, $CH_3Br$ and $CH_3I$ (which were consumed) and for $CHCl_3$ and $CHBr_3$ (which were emitted). Consumption rates of $CH_3Cl$ and $CH_3Br$ and emission rates of $CHCl_3$ from tundra and cyanobacterial mat sites were within the ranges reported from older and more established Arctic tundra elsewhere. Rough calculations showed total emissions and consumptions of these gases across the Arctic were small relative to other sources and sinks due to the small surface area represented by glacier forefields. We have demonstrated that glacier forefields can consume and emit halocarbons despite their young age and low soil development, particularly when cyanobacterial mats are present."

**(2) Were there any lab tests of potential impacts on storing the air samples in the vials or bags prior to analysis?**

The bags used for collection of the halocarbon samples were tested prior to use by flushing (three times) and then filling with the standard followed by storage in the same conditions as the sample bags for later analysis. Small changes were detected but are negligible compared to the overall changes measured in the chambers: the average change (n = 4) over 20 hours (maximum time between sampling and analysis) was 0.002 nmol $CH_3Cl$, -0.00001 nmol $CH_3Br$, 0.00001 $CH_3I$, 0.001 nmol $CHCl_3$, 0.00002 $CHBr_3$, 0.00001 $CH_2Br_2$.

Tests were not done on the exetainer vials for this study. However, a study by Faust and Liebig (2018) measured no significant changes in $CH_4$ and $CO_2$ concentrations for 15 mL exetainers over 28 days when stored at +4 ºC. The next time-point at which the exetainers were tested was after 84 days with concentrations for $CO_2$ and $CH_4$ found to be 0.6-14.4 % lower and up to 22% higher, respectively. However, the authors found that all exetainers after 84 days had a 'dent' in the septa.

Here, the samples stored in exetainers were analysed between 7 and 36 days after sampling. Although 32 of 150 samples in this study were analysed after more than the 28 days, they were stored for significantly less time than the 84 day period used in the Faust and Liebig (2018) study. Additionally, no 'dents' in the septa was not found in any exetainers analysed in this study. Therefore, it is reasonable to assume that storage up to 36 days in this study had negligible impacts on the gas concentrations inside the exetainers.

The following changes have been made in the manuscript to address the above:

Page 5, lines 27-29: "Exetainers were stored (within 4 hours of sampling) and transported at +4 °C until analysis in the UK within 36 days. Exetainers have previously been shown to be suitable for storage of $CO_2$ and $CH_4$ for at least 28 days, but not as long as 84 days (Faust and Liebig, 2018), and therefore we consider the storage time of up to 36 days to have had minimal impact on the measured concentrations."

Page 6, line 18-21: "Tests conducted on the sample bags found detectable but very small changes in gas

concentrations 20 hours after being flushed with the standard (+0.002 nmol $CH_3Cl$, -0.00001 nmol $CH_3Br$, +0.00001 $CH_3I$, +0.001 nmol $CHCl_3$, +0.00002 nmol $CHBr_3$, +0.00001 nmol $CH_2Br_2$)."

**(3) On p. 4, line 19 the radiocarbon age is 1850-1926 for the tundra site, but the abstract says approximately 1950 year old tundra.**

The tundra age of "approximately 1950 years old" was calculated as the age from the present day (specifically, the date of the fieldwork, 2017), i.e. the radiocarbon age, which is dated from 1 January 1950, plus the difference between 1950 and 2017. This was to keep the approximate ages of all land surfaces consistent with each other, as some of the surfaces are younger than the year 1950 and therefore have ages estimated from the year fieldwork was conducted. The age from today (rather than from BP) of the tundra would be 1917-1993 years old, which we approximated at 1950 years old. This has been clarified in the text on page 4, line 19-20:

"Radiocarbon dating near site *tundra* (~70 m west) has provided a date of exposure of 1850-1926 BP (Before Present, defined as $1^{st}$ January 1950 by the radiocarbon age scale; Hodkinson et al., 2003). This is equivalent to 1917-1993 years older (or approximately 1950 years) than the year of analysis (2017)."

---

## Author Response (AR2)

*Editors comment:*
Are the changes in the sampling bags really in absolute moles? Or should this rather by nmol/mol, i.e. ppt? Please check this

*Authors response:*
We have checked this and the changes are in absolute moles, this has been clarified in the text.

[revised manuscript text omitted]